# Learning to Rank Context for Named Entity Recognition Using a Synthetic Dataset

**Arthur Amalvy**
Laboratoire Informatique d'Avignon
arthur.amalvy@univ-avignon.fr

**Vincent Labatut**
Laboratoire Informatique d'Avignon
vincent.labatut@univ-avignon.fr

**Richard Dufour**
Laboratoire des Sciences du Numérique de Nantes
richard.dufour@univ-nantes.fr

## Abstract

While recent pre-trained transformer-based models can perform named entity recognition (NER) with great accuracy, their limited range remains an issue when applied to long documents such as whole novels. To alleviate this issue, a solution is to retrieve relevant context at the document level. Unfortunately, the lack of supervision for such a task means one may have to settle for unsupervised approaches. Instead, we propose to generate a synthetic context retrieval training dataset using Alpaca, an instruction-tuned large language model (LLM). Using this dataset, we train a neural context retriever based on a BERT model that is able to find relevant context for NER. We show that our method outperforms several unsupervised retrieval baselines for the NER task on an English literary dataset composed of the first chapter of 40 books, and that it performs on par with re-rankers trained on manually annotated data, or even better.

## 1 Introduction

Named entity recognition (NER), as a fundamental building block of other natural language processing (NLP) tasks, has been the subject of a lot of researchers' attention. While pre-trained transformer-based models are able to solve the task with a very high F1-score (Devlin et al., 2019; Yamada et al., 2020), they still suffer from a range limitation caused by the quadratic complexity of the attention mechanism in the input sequence length. When applied to long documents such as entire novels, this range limitation prevents models from using global document-level context, since documents cannot be processed as a whole. However, document-level context is useful for entity disambiguation: for NER, lacking access to this context results in performance loss (Amalvy et al., 2023). To try and solve the range limitation of transformers, a number of authors recently explored "efficient transformers", with a sub-quadratic complexity (see Tay et al. (2022) for a survey). Still, long sequences remain challenging (Tay et al., 2021).

Retrieval-based methods can circumvent these issues, by retrieving relevant context from the document and concatenating it to the input. Unfortunately, no context retrieval dataset is available for NER, and manually annotating such a dataset would be costly. This lack of data prevents from using a specialized supervised context retrieval method.

Meanwhile, instruction learning has very recently seen an increasing focus from the NLP community (Renze et al., 2023). Instructions-tuned models are able to solve a variety of tasks (classification, regression, recommendation...) provided they can be formulated as text generation problems. Inspired by the progress of these models, we propose to generate a synthetic context retrieval dataset tailored to the NER task using Alpaca (Taori et al., 2023), a recently released instruction-tuned large language model (LLM). Using this synthetic dataset, we train a neural context retriever that ranks contexts according to their relevance to a given input text for the NER task. Since applying this retriever on a whole document is costly, we instead use it as a re-ranker, by first retrieving a set of candidate contexts using simple retrieval heuristics and then keeping the best sentences among these.

In this article, we study the application of our proposed method on a corpus of literary novels, since the length of novels make them particularly prone to the range limitation of transformers. We first describe the generation process of our synthetic retrieval dataset. We then evaluate the influence of the neural context retriever trained on this dataset in terms of NER performance, by comparing it to unsupervised retrieval methods. We also compare our re-ranker with other supervised re-rankers trained on a manually annotated context retrieval dataset unrelated to literary NER. We

explore whether the number of parameters of the LLM used for generating our synthetic context retrieval dataset has an influence on the trained neural retriever by comparing Alpaca-7b and Alpaca-13b, and the influence of the number of candidate sentences to retrieve before re-ranking. Additionally, we study the influence of the context window (i.e. context retrieval range) on the performance of the baseline retrieval heuristics and of our neural context retriever. We release our synthetic dataset and our code under a free license[1].

## 2 Related Work

### 2.1 Re-Rankers

In the field of information retrieval, re-rankers are used to complement more traditional retrievers such as BM25 (Robertson, 1994). In that setting, given a query and a set of passages, a traditional retriever is first used to retrieve a certain number of candidate passages. Then, a more computationally expensive re-ranker such as MonoBERT (Nogueira and Cho, 2020) or MonoT5 (Nogueira et al., 2020) retrieves the top-$k$ passages among these candidates. While supervised re-rankers offer good performance, they necessitate training data, but annotation can be costly. To avoid this issue, we propose a re-ranker tailored to the NER task that is trained on a synthetic retrieval dataset.

### 2.2 Named Entity Recognition and Context Retrieval

Different external context retrieval techniques have been leveraged for NER (Luo et al., 2015; Wang et al., 2021; Zhang et al., 2022). Comparatively, few studies focus on document-level context retrieval, which can be used even when no external resources are available. Luoma and Pyysalo (2020) introduce majority voting to combine predictions made with different contexts, but their study is restricted to neighboring sentences and ignores document-level context.

Recently, Amalvy et al. (2023) explore the role of global document-level context in NER. However, their study only considers simple unsupervised retrieval heuristics due to the lack of supervision of the context retrieval task. By contrast, we propose to solve the supervision problem by generating a synthetic context retrieval dataset using a LLM. This allows us to train a neural model that outperforms unsupervised heuristics.

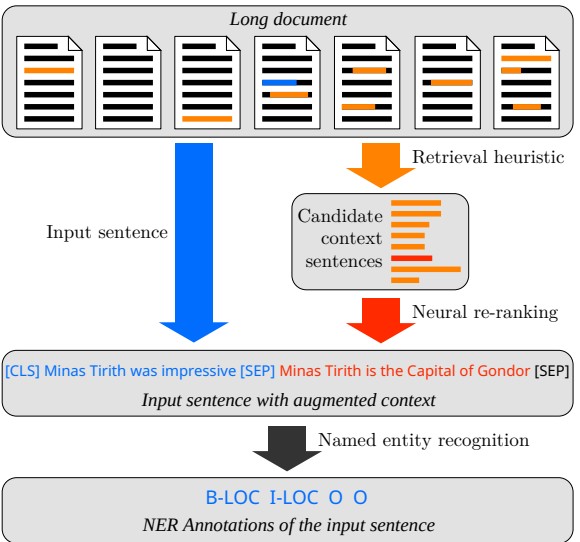

Figure 1: Overview of our neural re-ranker performing context retrieval.

### 2.3 Instruction-Following Large Language Models

Instruction-following LLMs are trained to output text by following user instructions. Multi-task learning (Wei et al., 2022; Sanh et al., 2022; Muennighoff et al., 2023) and fine-tuning on instructions datasets (Taori et al., 2023; Chiang et al., 2023) are two common training paradigms for these models. One of the goals of these training methods is to obtain good zero-shot performance for a variety of tasks, making instruction-following models very versatile.

Our synthetic dataset generation task calls for models producing longer outputs than multi-task learning based models such as BloomZ or mT0, which are biased in favor of short outputs (Muennighoff et al., 2023). We therefore select Alpaca (Taori et al., 2023), an instruction-tuned model based on Llama (Touvron et al., 2023).

## 3 Method

In this section, we start by introducing the *document-level context retrieval* task for NER. Then, since no dataset exists for this task, we detail how we generate a synthetic context retrieval dataset using an instruction-following LLM in Section 3.2. We describe the neural context retriever that we train using this dataset in Section 3.3. Figure 1 provides an overview of the process we propose to retrieve context with our neural retriever. Finally, we provide details on our dataset in Section 3.4.

---

[1] https://github.com/CompNet/conivel/tree/gen

| **Description** (all classes) | **Prompt template** '`{INPUT SENTENCE}`' `- In the preceding sentence, {ENTITY} is a character. Invent a one-sentence description for this character, mentioning their name.`
**Example input sentence** *[One-Eye]'s handicap in no way impairs his marvelous hindsight.*
**Example generated context** *One-Eye is a wise and mysterious character with a penchant for coming up with invaluable insights after the fact.* |
|---|---|
| **Action** (`PER` only) | **Prompt template** `Invent a single sentence depicting the character '{ENTITY}' performing an action, mentioning their name.`
**Example input sentence** *"It's my stomach, [Croaker],"*
**Example generated context** *Croaker was whistling a jaunty tune as he strolled through the Park.* |
| **Movement** (`LOC` only) | **Prompt template** `Invent a single sentence depicting a character of your invention going to {ENTITY}. You must mention the name of the character.`
**Example input sentence** *Lightning from a clear sky smote the [Necropolitan Hill].*
**Example generated context** *The gothic vampire Count Necropolis ventured to Necropolitan Hill, his ancient stomping grounds.* |

Table 1: Prompts templates and examples of positive context retrieval samples generated by Alpaca-7b.

| **Positive examples swapping** | **Example input sentence** *We left in pretty good time and came after nightfall to Klausenburgh*
**Example context** *Forley was an adventurous and daring individual who was never afraid to take risks* |
|---|---|
| **Negative sampling** | **Example input sentence** *I am afraid that I have been tempted into too great length about the Italian Catherine; but in truth she has been my favourite.*
**Example context** *said Alice, as she swam about, trying to find her way out.* |

Table 2: Examples of negative context retrieval samples generated by *positive examples swapping* or *negative sampling*.

## 3.1 Document-Level Context Retrieval

We define the document-level context retrieval problem as follows: given a sentence $s_i$ and its enclosing document $D_i$, we must retrieve $S_i$, a set of $k$ relevant sentences in $D_i$. We define relevance as being helpful for predicting the entities in $s_i$ by allowing entity class disambiguation. After retrieving $S_i$, we concatenate its sentences to $s_i$, in order to form a list $\hat{S}_i$, while keeping the relative ordering of sentences in $D_i$.

After context retrieval, we compute NER labels for sentence $s_i$ by inputting $\hat{S}_i$ into a NER model and keeping only the labels related to $s_i$.

## 3.2 Context Retrieval Dataset Generation

Unfortunately, no context retrieval dataset exists for the task of retrieving relevant context for NER at the document level. Without supervision, we cannot train a supervised model to solve this task and are restricted to unsupervised retrieval methods. Therefore, we set out to generate a synthetic dataset using Alpaca (Taori et al., 2023).

We define a context retrieval dataset as a set of 3-tuples $(s_i, s_j, y)$, where:

- $s_i$ is the input sentence (the sentence for which we wish to retrieve context).

- $s_j$ is the retrieved context sentence.

- $y$ is the relevance of $s_j$ with respect to $s_i$:

either 1 if $s_j$ is relevant, or 0 otherwise.

Depending on whether the example is positive ($y = 1$) or negative ($y = 0$), we design different generation methods.

**Positive Examples**  For each NER class in our dataset (PER (person), LOC (location) and ORG (organization)), we empirically observe which types of sentences can help for disambiguation with other classes. We determine that the following categories of sentences are useful:

- For all classes (PER, LOC and ORG): descriptions of the entity explicitly mentioning it (**Description**).

- For the PER class only: sentences describing the PER entity performing an action (**Action**).

- For the LOC class only: sentences describing a PER entity going to the LOC entity (**Movement**).

We design a prompt for each of these types of sentences that, given an input sentence and an entity, can instruct Alpaca to generate a sentence of this type. Table 1 show these prompts and some example sentences generated by Alpaca-7b. Given a sentence $s_i$ from our NER training dataset, we select an entity from $s_i$ and generate $s_j$ by instructing Alpaca with one of our prompts. This allows us to obtain a positive context retrieval example $(s_i, s_j, 1)$. We repeat this process on the whole NER training dataset. To avoid overfitting on the most frequent entities, we generate exactly one example per unique entity string in the dataset. We filter sentences $s_j$ that do not contain the target entity string from the input sentence $s_i$.

**Negative Examples**  We use two different techniques to generate negative examples:

- *Negative sampling*: given a sentence $s_i$ from a document $D_i$, we sample an irrelevant sentence by randomly selecting a sentence from another document $D_j$. Negative sampling generates some contexts that contain entities, and some contexts that do not.

- *Positive examples swapping*: given an existing positive example $(s_i, s_j, 1)$, we generate a negative example $(s_i, s_k, 0)$ by replacing $s_j$ with a context $s_k$ from another positive example. Since we generate a single positive example per entity string, $s_k$ has a high chance

not to contain an entity from $s_i$. Note that all context sentences $s_k$ contain an entity. We use this additional generation technique because we found that generating examples using only negative sampling led to the model overfitting on contexts containing entities (since negative sampling does not guarantee that the retrieved context contains an entity).

**Generated Dataset**  We generate two context retrieval datasets: one with Alpaca-7b, and one with Alpaca-13b. They contain respectively 2,716 and 2,722 examples[2]. Interestingly, the models display a knowledge of some entities from the dataset when generating samples, probably thanks to their pre-training. For example, when asked to generate a context sentence regarding *Gandalf the Grey* from *The Lords of the Rings*, an Alpaca model incorporates the fact that he is a wizard in the generated context even though it does not have this information from the input text.

### 3.3 Neural Context Retriever

For our neural context retriever, we use a BERT model (Devlin et al., 2019) followed by a regression head. For a given sentence and a candidate context, the retriever outputs the estimated relevance of the candidate context between 0 and 1. Because applying our model on the whole document is computationally costly, we use our retriever as a re-ranker: we first retrieve some *candidate contexts* using simple heuristics (cf. Section 4.2) before re-ranking these sentences.

### 3.4 Dataset

We use the English NER literary dataset constituted by Dekker et al. (2019) and then corrected and enhanced by Amalvy et al. (2023). We chose this dataset specifically for the length of its documents. It is composed of the first chapter of 40 novels, and has 3 NER classes: Person name (PER), Location (LOC) and Organization (ORG). We split each document in sentences. To perform context retrieval on the entire novels, we also collect the full text of each novel.

---

[2]The number of examples between the two datasets is different because of the filtering process when generating positive examples. This filtering occurred very rarely, as the vast majority of generated sentences included the target entity.

## 4 Experiments

### 4.1 Unsupervised Retrieval Baselines

We compare the performance of our neural context retriever with the following unsupervised baselines from Amalvy et al. (2023):

- `no retrieval`: The model performs the NER task on each sentence separately, without additional context.

- `surrounding`: Retrieves sentences that are just before and after the input sentence.

- `bm25`: Retrieves the $k$ most similar sentences according to BM25 (Robertson, 1994).

- `samenoun`: Retrieves $k$ sentences that contain at least a noun in common with the input sentence. We identify nouns using NLTK (Bird et al., 2009) as done by Amalvy et al. (2023).

### 4.2 Inference

Contrary to Amalvy et al. (2023), we retrieve context from the whole document instead of the first chapter only. We use our neural context retriever as a re-ranker. For each sentence $s_i$ of the dataset, we first retrieve a total of $4n$ *candidate context sentences* using the following heuristics:

- $n$ sentences using the `bm25` heuristic.

- $n$ sentences using the `samenoun` heuristic.

- The $n$ sentences that are just before the current sentence.

- The $n$ sentences that are just after the current sentence.

Note that the same candidate sentence can be retrieved by different heuristics: to avoid redundancy, we filter out repeated candidates. We experiment with the following values of $n : \{4, 8, 12, 16, 24\}$.

After retrieving $4n$ candidate contexts, we compute their estimated relevance with respect to the input sentence using our neural context retriever. We then concatenate the top-$k$ contexts to the input sentence $s_i$ before performing NER prediction. We report results for a *number of retrieved sentences k* from 1 to 8[3].

### 4.3 Re-ranker Baselines

We compare our system with the following trivial re-rankers baselines:

- `random re-ranker`: We retrieve $4n$ sentences with all the unsupervised heuristics (as described in Section 4.2), and then randomly select $k$ sentences from these candidates.

- `bucket random re-ranker`: Same as above, except we select $k/4$ sentences for each heuristic.

These baselines allow us to observe the real benefits of re-rankers. We also compare our retriever with supervised re-rankers trained on the MSMarco passage retrieval dataset (Bajaj et al., 2018):

- `bm25+monobert`: We retrieve 16 sentences using BM25, and then use the MonoBERT-large (Nogueira and Cho, 2020) re-ranker to retrieve the $k$ most relevant sentences. This configuration simulates a classic information retrieval setup.

- `all+monobert`: We retrieve $4n$ sentences using all the unsupervised heuristics (as described in Section 4.2), and then use the MonoBERT-large re-ranker to retrieve the $k$ most relevant sentences. This configuration is the closest to our neural retriever, so we use it to compare the influence of the training dataset.

- `bm25+monot5`: Same as `bm25+monobert`, but using a MonoT5 reranker (Nogueira et al., 2020).

- `all+monot5`: Same as `all+monobert`, but using a MonoT5 re-ranker.

### 4.4 Training

We train the neural context retriever on the synthetic dataset we generated as described in Section 3.2 for 3 epochs with a learning rate of $2 \times 10^{-5}$. To study the importance of the LLM size that we use when generating our synthetic dataset, we generate our context retrieval dataset with two differently sized versions of the same model; as detailed in Section 3.2: Alpaca-7B (7 billion parameters) and Alpaca-13B (13 billion parameters) (Taori et al., 2023).

---

[3]Note that the `samenoun` heuristic is an exception, as it may retrieve fewer than $k$ sentences. This is because some input sentences have fewer than $k$ contexts in the document with at least one common noun.

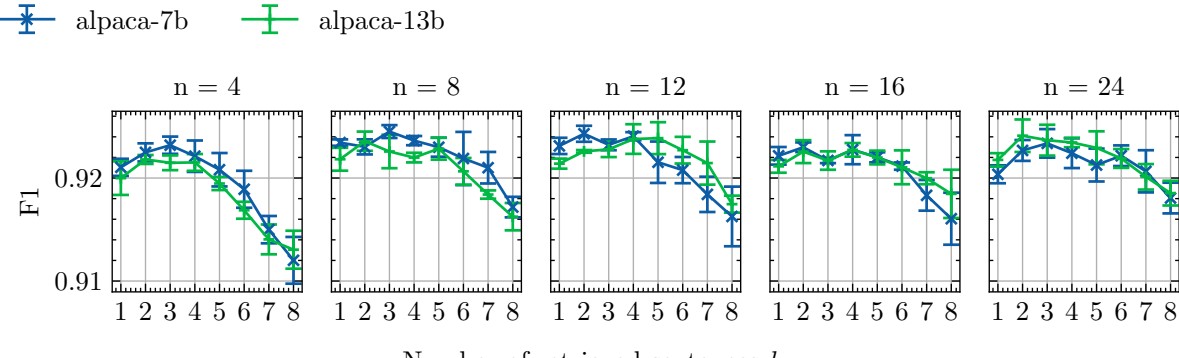

Figure 2: Effect of the number of candidate sentences $4n$ on the performance of our neural context retriever, trained using a dataset generated with Alpaca-7b or Alpaca-13b.

## 4.5 Named Entity Recognition Model

In all of our experiments, we use a pretrained BERT model (Devlin et al., 2019) followed by a classification head fine-tuned for 2 epochs with a learning rate of $2 \times 10^{-5}$. We use the `bert-base-cased` checkpoint from the `huggingface transformers` library (Wolf et al., 2020).

## 4.6 Evaluation

We compute the F1-score of our different configurations using the default mode of the `seqeval` (Nakayama, 2018) library for reproducibility. We perform all experiments 5-folds, and report the mean of each metric on the 5 folds. To compare the different retrieval configurations fairly, we train a single NER model and compare results using this model and different retrieval methods. For retrieval methods that are not deterministic (some re-rankers and the `samenoun` heuristic[4]), we report the mean of 3 runs to account for variations between runs.

## 5 Results

### 5.1 Neural Re-Ranker Configuration Selection

In this section, we set out to find the optimal configuration of our neural context retriever. We survey the number of candidate contexts $4n$ to retrieve before re-ranking, and the size of the model. Figure 2 shows the NER performance of the neural retriever trained with either Alpaca-7b or Alpaca-13b for different values of $n$.

**Number of candidate contexts** We find that $n = 4$ is too low, probably because not enough relevant candidate contexts are retrieved to obtain good performance. Increasing $n$ leads to better performance for high values of the number of retrieved sentences $k$. However, the best performance is obtained for values of $k$ between 2 and 5.

**LLM model size** We find that the size of the Alpaca model (7b versus 13b) has very little influence on the final NER performance for all surveyed values of $n$.

In the rest of this article, we discuss the best configuration we found in this experiment unless specified otherwise: the neural retriever trained using Alpaca-7b and with $n = 8$ and $k = 3$. We also keep the same configuration for the supervised baselines. With that configuration, our neural re-ranker has a F1 of 98.01 on the evaluation portion of our synthetic context retrieval dataset.

### 5.2 Comparison with Unsupervised Retrieval Methods

Figure 3 shows the F1 of our NER model when using our re-ranker or different unsupervised retrieval methods. Retrieval with any method is beneficial (except when retrieving 6 sentences or more with `surrounding` or `bm25`), but our neural re-ranker generally outperforms all the simple heuristics. Specifically, it beats the `no retrieval` configuration by around 1 F1 point, with its peak performance being when the number of retrieved sentences $k = 3$.

**Per-book results** Figure 4 shows the F1 score per book for our retrieval method and the unsupervised retrieval methods, for a single retrieved sentence. Our neural context retriever is better or on-par with

---

[4]The `samenoun` method is not deterministic since it randomly retrieves among candidate sentences that have a common noun with the input sentence.

| Input sentence | Candidate contexts | Score |
|---|---|---|
| *"The Ministry of Truth – Minitrue, in Newspeak [Newspeak was the official language of Oceania."* | *"The Ministry of Truth, which concerned itself with news, entertainment, education, and the fine arts."* | 1.0 |
| | *"Winston made for the stairs."* | 0.0 |
| *"The eldest of these, and Bilbo's favourite, was young Frodo Baggins."* | *"Then he disappeared inside with Bilbo, and the door was shut."* | 1.0 |
| | *I feel I need a holiday, a very long holiday, as I have told you before.* | 0.0 |

Table 3: Example predictions of the neural context retriever.

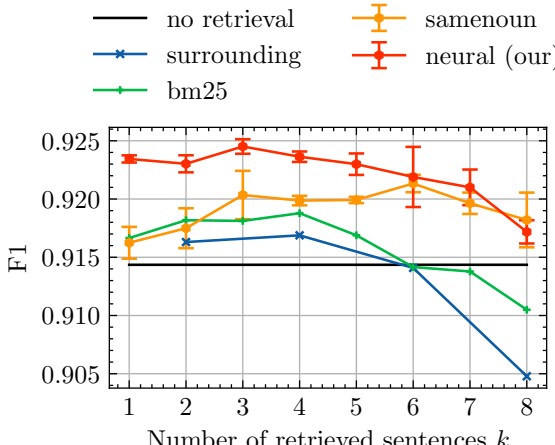

Figure 3: NER F1 score comparison between our neural retriever and unsupervised retrieval methods.

other configurations for 50% of the novels (20 out of 40). Performance varies a lot depending on the novel, with the highest improvement of the neural method over the `no retrieval` configuration being 8.1 F1 points for *The Black Company*.

### 5.3 Comparison with Re-Rankers

We compare the F1 of our neural re-ranker with other re-rankers trained on MSMarco (Bajaj et al., 2018) in Figure 5. All the supervised methods outperforms the random re-rankers, showing that they can indeed retrieve useful context. Our model outperforms the other supervised methods, albeit slightly. While results are close, this is still interesting as we used generated data only while the other supervised approaches were trained with gold annotated data. We note that using multiple pre-retrieval heuristics (the `all+` configurations and our neural re-ranker) is generally better than using only BM25, which tends to show that retrieving sentences with BM25 alone may lead to missing important context sentences.

### 5.4 Size of the Context Window

To understand the role of the size of the *context window*, we compare between retrieving context in the first chapter of each book only as in (Amalvy et al., 2023) and retrieving context on the entire book. Figure 6 shows the performance of the `bm25`, `samenoun` and `neural` configurations depending on the context window. For the `neural` configuration, since the number of retrieved sentences $4n$ may have an influence on the results, we report performance with values of $n$ in $\{4, 8, 12\}$.

**bm25** Retrieving a few sentences from the current chapter seems more beneficial to the `bm25` heuristic. However, performance decreases sharply when retrieving more than 2 sentences in the same chapter. This effect is weaker when retrieving context in the whole book, which seems indicative of a saturation issue, where the number of helpful sentences that can be retrieved using `bm25` in a single chapter is low.

**samenoun** The performance of the `samenoun` heuristic seems to follow the same pattern as `bm25`. Retrieving a few sentences from the current chapter seems better than doing so in the full document. This might be because the current chapter has a higher chance of containing sentences talking about an entity of the input sentence. Meanwhile, it is better to retrieve contexts in the whole book for values of $k > 4$, possibly because such a number of retrieved sentences means the heuristic has a high enough chance of retrieving a relevant sentence at the book level.

**neural** The context window seems critical for the performance of the neural retriever. Retrieving context only in the current chapter suffers from a large performance drop compared to retrieving context in the whole book. This is true even for different

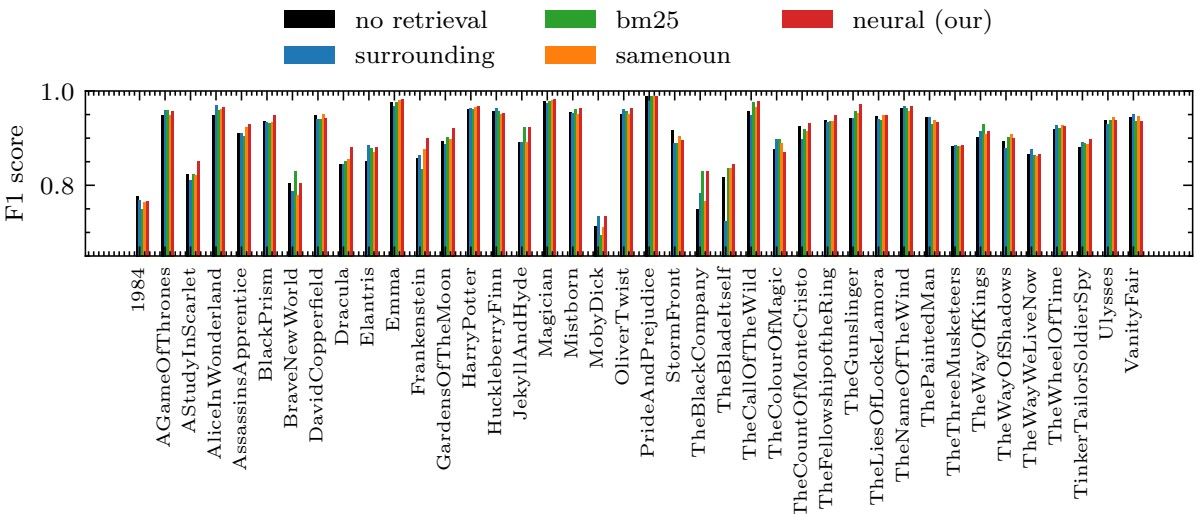

Figure 4: NER F1 Score per book for different retrieval methods, with a number of retrieved sentences $k = 1$.

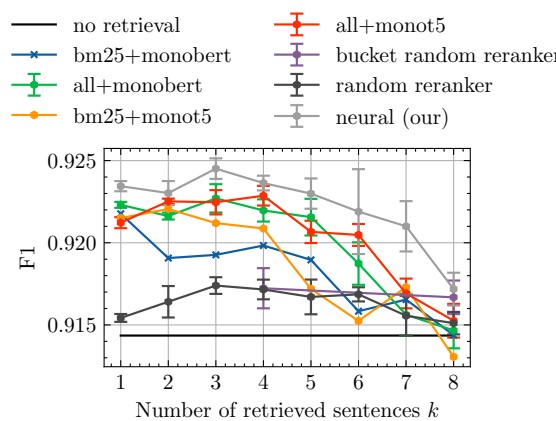

Figure 5: NER F1 score comparison between our retriever and different re-ranking methods.

values of the number of retrieved sentences $4n$.

## 6 Conclusion

In this article, we proposed to train a neural context retriever tailored for NER on a synthetic context retrieval dataset. Our retrieval system can be used to enhance the performance of a NER model, achieving a gain of around 1 F1 point over a raw NER model and beating unsupervised retrieval heuristics. We also demonstrated results on par or better than re-rankers trained using manually annotated data.

As stated in Section 7, a limitation of our neural context retriever is that the relevance of each candidate context sentence is estimated separately. To account for potential interactions, we could iteratively add each candidate to the input sentence and rank the remaining candidates with respect to

that newly formed input. Future works may expand on this article by exploring whether taking these interactions into account can have a positive impact on performance.

We also note that our proposed method, generating a retrieval dataset using an instructions-tuned LLM, could be leveraged for other NLP tasks where global document-level retrieval is useful. The only pre needed to generate such a dataset is the knowledge of which types of context samples can help to improve the performance for a local task, in order to generate positive samples. Future works may therefore focus on applying this principle on different tasks.

## 7 Limitations

### 7.1 Candidate Contexts' Interactions

Our proposed neural context retriever does not consider the relations between retrieved sentences. This can lead to at least two issues:

- When the model is asked to retrieve several context sentences, it can return sentences with redundant information. This makes the input of the NER task larger, and therefore the inference slower, for no benefit.

- If an input sentence contains two entities that need disambiguation, the model might retrieve several examples related to one of the entities only.

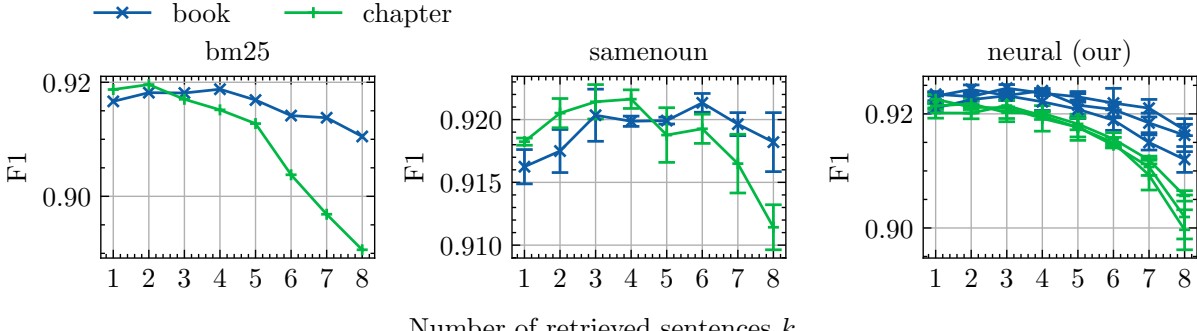

Figure 6: NER performance of different retrieval methods for different context window sizes as a function of the number of retrieved sentences $k$. For the neural configuration, we report performance for values of $n$ in $\{4, 8, 12\}$.

## 7.2 Computational Costs

Our context retrieval approach extends the range of transformer-based models by avoiding the quadratic complexity of attention in the sequence length. However, retrieval in itself still bears a computational cost that increases linearly in the number of sentences of the document in which it is performed. Additionally, retrieving more context sentences also increases the computational cost of predicting NER labels.

## 7.3 Biases

LLMs are prone to reproducing biases found in the training data (Sheng et al., 2019). The original Llama models, which are used as the basis for fine-tuning Alpaca models, display a range of biases (about gender, religion, age...) (Touvron et al., 2023) that could bias the context retrieval dataset. This, in turn, could bias the neural context retriever model trained on that dataset.

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

# A Comparison with Unsupervised Retrieval Methods: Precision/Recall Breakdown

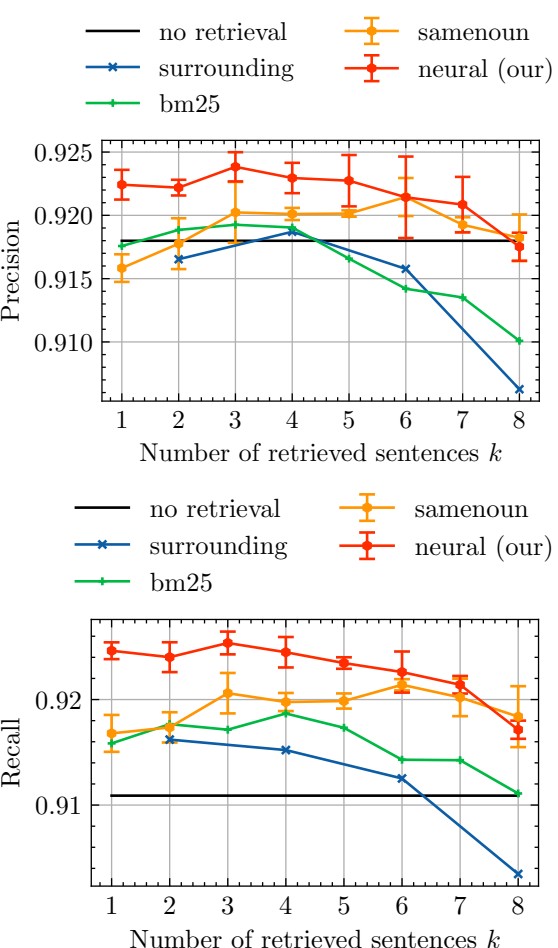

Figure 7: NER Precision (top) and Recall (bottom) comparison between our neural retriever and unsupervised retrieval methods.

As seen in the bottom plot of Figure 7, unsupervised retrieval methods and our neural re-ranker

all benefits recall (except `surrounding` when the number of retrieved sentences $k = 8$), but, as observed in the top plot, they can all have a negative impact on precision depending on the value of $k$.