# OpenReview forum: "Learning to Rank Context for Named Entity Recognition Using a Synthetic Dataset"
_EMNLP/2023/Conference — EMNLP 2023 Main_

### Official Review · Reviewer_NDUN · 2023-07-31

**Soundness:** 4

**Excitement:**

3: Ambivalent: It has merits (e.g., it reports state-of-the-art results, the idea is nice), but there are key weaknesses (e.g., it describes incremental work), and it can significantly benefit from another round of revision. However, I won't object to accepting it if my co-reviewers champion it.

**Paper Topic And Main Contributions:**

This paper builds on the work of Amalvy et al. (2023) on retrieval-based context augmentation for document-level NER in novels. The authors propose to use instruction-following LLMs like Alpaca to generate sentences with given entities in them. Generation is guided by engineering specific prompts using a sentence including the given entity. The dataset, which also contains negative examples, is used to train a neural regression model that assigns a relevance score to two input sentences. This model is then used to rerank sentences retrieved by some heuristic from the input document in order to augment the context of the NER model. The authors show that this approach results in improvement of NER quality measure by 1 point increase in F1 on Dekker et al. (2019) dataset of novels.

**Questions For The Authors:**

* Can you elaborate on why training on negative sampling only leads to "overfitting on context containing entities", claimed on line 215?
* In positive sampling, I don't understand the need to filter out sentences $s_j$ that do not contain the target entity (line 198). Why not continue generation until the target entity appears? Is this related to the fact that the datasets generated with Alpaca 7B and 13B are of different sizes? This fact is also unclear to me.
* More details about training of the neural reranker should be given in the paper (line 276). Was the model evaluated intrinsically (on a portion of the synthetic dataset)?
* How do you identify nouns in the "samenoun" setup? Do you use NLTK as done by Amalvy et al. (2023)?
* I'm not sure if the claim on line 288 is true. My understanding is that the method of Amalvy et al. (2023) can perform global retrieval on the whole document and doesn't "need NER annotations". Extension to whole novels is therefore straightforward and unrelated to NER. Please elaborate on this to remove any confusion.
* The hypothesis on line 406 should be tested empirically.

**Reasons To Accept:**

* Up to my knowledge, this is a novel use of LLMs in dataset generation for NER.
* The provided anonymous repository is well organized which facilitates reproduction.

**Reasons To Reject:**

* Dataset generation is described for specific entity types (PER, LOC, ORG). Because of the dependence on LLM prompts, it is not guaranteed that its extension to new types will result in similar performance gain.
* It is not clear from the paper if the synthesized dataset allows for training of trainers that generalize to other NER tasks or if new datasets should be generated every time the approach is used. This is computationally expensive and would affect its adoption.
* In order to better understand the importance of the reranker, a random baseline should be included which select $k$ random sentences from the $4n$. Also compare with randomly selecting $k/4$ sentences from each bucket of $n$.
* A missing baseline to assess the relevance of the dataset in my opinion is to compare with a neural reranker trained on any available similarity dataset.
* Qualitative analysis is needed to understand where the increase of 1 point of F1 is coming from and if it justifies the computational resources needed to implement the method.
* The proposed method to generate the dataset is completely unrelated to the document from which the prompt sentence is selected. It is also unrelated to the output of the retrieval heuristic on which the reranker operates. It might be better to use the LLMs to sort through the output of the heuristics. This creates a discrepancy between train and test distributions that might be harmful for the reranker. I understand though that performance gains in NER as a result of using this dataset indicates that the reranker is still able to function with the test distribution. An exploration of this question is a welcomed addition in my opinion.
* Some points in the method description need clarification (see the remarks below).

**Reproducibility:**

4: Could mostly reproduce the results, but there may be some variation because of sample variance or minor variations in their interpretation of the protocol or method.

**Reviewer Confidence:**

4: Quite sure. I tried to check the important points carefully. It's unlikely, though conceivable, that I missed something that should affect my ratings.

---

> ### Author Rebuttal · Authors · 2023-08-29
>
> Dear reviewer,
>
> Thank you for your review! We tried to answer to your remarks as best as we could below:
>
>
> > Dataset generation is described for specific entity types (PER, LOC, ORG). Because of the dependence on LLM prompts, it is not guaranteed that its extension to new types will result in similar performance gain.
>
> We worked on ENAMEX entities, which are among the most common entities types, but we agree that it would be interesting to study the effect of the method on other entity types. However, these entity types should be chosen accordingly, since the method might be irrelevant for some of these (dates, amounts...).
>
>
> > It is not clear from the paper if the synthesized dataset allows for training of trainers that generalize to other NER tasks or if new datasets should be generated every time the approach is used. This is computationally expensive and would affect its adoption.
>
> This is an interesting question, that would be an interesting extension to our work. However, we are concerned by the lack of long-range NER datasets for evaluation.
>
>
> > In order to better understand the importance of the reranker, a random baseline should be included which select $k$ random sentences from the $4n$. Also compare with randomly selecting sentences from each bucket of $n$.
>
> This is a fair point. To address it, we performed your first suggested experiment with $n=8$. Here are the results we observed (f1, mean of 3 runs):
>
> | k |     1 |     2 |     3 |     4 |     5 |     6 |     7 |     8 |
> | - | ----- | ----- | ----- | ----- | ----- | ----- | ----- | ----- |
> |   | 91.57 | 91.64 | 91.74 | 91.72 | 91.67 | 91.69 | 91.56 | 91.51 |
>
> The results are lower that using our reranker (or the MonoBERT reranker, see below) for all values of $k$, highlighting the added benefits of trained rerankers. We will include this experiment in a revised version of the article, as well as your second suggested experiment.
>
>
> > A missing baseline to assess the relevance of the dataset in my opinion is to compare with a neural reranker trained on any available similarity dataset.
>
> We performed additional experiments on other baselines to answer this concern. We compared our re-ranker to a pretrained MonoBERT re-ranker (Nogueira and Cho, 2019). We chose MonoBERT since we wanted to compare to a model with a similar architecture in order to study the difference of performance caused by the training dataset. Still, note that the version of MonoBERT we employed is based on BERT large (huggingface checkpoint: `castorini/monobert-large-msmarco`), while our model is based on BERT base.
>
> We experimented with two variants of a MonoBERT based system:
>
> - bm25 -> MonoBERT, to replicate a classic BM25 + reranker baseline
> - [bm25, samenoun, surrounding] -> MonoBERT with $n=8$, to compare with the neural setup of our article (as in paragraph Inference of Section 4.1)
>
> We obtained the following mean results on 3 runs (F1-score):
>
> | System             /                 k    |     1 |     2 |     3 |     4 |     5 |     6 |     7 |     8 |
> | ----------------------------------------- | ----- | ----- | ----- | ----- | ----- | ----- | ----- | ----- |
> | bm25 -> MonoBERT                          | 92.18 | 91.91 | 91.93 | 91.98 | 91.90 | 91.58 | 91.65 | 91.44 |
> | [bm25, samenoun, surrounding] -> MonoBERT | 92.23 | 92.16 | 92.27 | 92.20 | 92.16 | 91.87 | 91.56 | 91.47 |
> | our                                       | 92.34 | 92.30 | 92.45 | 92.36 | 92.30 | 92.19 | 92.10 | 91.72 |
>
> Overall, our model obtains better performance for all values of $k$, even with MonoBERT having more parameters than our model. Our hypothesis is that the performance differences come from our retrieval dataset being specifically suited for NER and for the domain at hand. We will include these results in a revised version of the article.
>
>
> > Qualitative analysis is needed to understand where the increase of 1 point of F1 is coming from and if it justifies the computational resources needed to implement the method.
>
> We did include a few examples of retrieved contexts in Table 3. In order to go further, we are currently manually checking the cases where the re-ranker corrected a NER prediction that was erroneous when using one of the retrieval heuristics.
>
>
> > The proposed method to generate the dataset is completely unrelated to the document from which the prompt sentence is selected. It is also unrelated to the output of the retrieval heuristic on which the reranker operates. It might be better to use the LLMs to sort through the output of the heuristics. This creates a discrepancy between train and test distributions that might be harmful for the reranker. I understand though that performance gains in NER as a result of using this dataset indicates that the reranker is still able to function with the test distribution. An exploration of this question is a welcomed addition in my opinion.
>
> If we understand your question correctly, you are asking if the discrepancy between the distribution of examples in the synthetic retrieval dataset and the distribution of sentences retrieved by the heuristics can have an impact on performance. As you point out, the results tend to show that the reranker is able to function on the distribution of retrieved sentences, but the discrepancy might still affect results negatively. This is an interesting question that warrants at least a mention in the article. An LLM might be able to sort through the output of the heuristics to produce retrieval examples, but we are concerned about the the correctness of the generated examples. We tested a similar approach where we generated positive examples by checking whether or not contexts retrieved by the heuristics could help the NER prediction, but the quality of the generated examples was subpar which made for a noisy retrieval dataset that did not lead to good retrieval performance in practice.
>
>
> > Can you elaborate on why training on negative sampling only leads to "overfitting on context containing entities", claimed on line 215?
>
> A large portion of contexts retrieved using negative sampling do not contain any entities (66.37\% of the sentences of our NER dataset do not contain any entities). We observed that, due to this, the model would overfit and mostly predict that a context containing an entity was relevant (and a context with no entity irrelevant). Adding irrelevant contexts with entities fixed that issue.
>
>
> > In positive sampling, I don't understand the need to filter out sentences that do not contain the target entity (line 198). Why not continue generation until the target entity appears?
>
> We could have adapted sampling to force the model to continue generating until the target entity is found (for example, by not allowing sampling the END token until the entity is generated). However, there is no guarantee that the model will ever generate the entity. This filtering only occurred in a very few cases, as the model generated a context containing the entity almost every time.
>
>
> > Is this related to the fact that the datasets generated with Alpaca 7B and 13B are of different sizes? This fact is also unclear to me.
>
> You are right, the (small) difference in size between both datasets is due to the filtering process. We added this precision to the paper to increase clarity.
>
>
> > More details about training of the neural reranker should be given in the paper (line 276). Was the model evaluated intrinsically (on a portion of the synthetic dataset)?
>
> To answer your question, we measured that the reranker has a F1 score of 98.01 on a synthetic test set generated 5-folds on the test portion of the NER dataset. The task on this synthetic train set may be easier than the task in real conditions, but the NER performance increase and retrieved contexts (Table 3) make us confident that the neural re-ranker is returning relevant contexts. We improved the article to add the reranker's score on the synthetic test set.
>
>
> > How do you identify nouns in the "samenoun" setup? Do you use NLTK as done by Amalvy et al. (2023)?
>
> We used the same method as Amalvy et al. (2023). We added this fact in the article for clarity.
>
>
> > I'm not sure if the claim on line 288 is true. My understanding is that the method of Amalvy et al. (2023) can perform global retrieval on the whole document and doesn't "need NER annotations". Extension to whole novels is therefore straightforward and unrelated to NER. Please elaborate on this to remove any confusion.
>
> You are right in saying that the method of Amalvy et al. can be extended to the whole document and do not need NER annotations. However, in their article, they only retrieved context on the first chapter of each book, which is the only chapter annotated in the book. By contrast, we applied their retrieval method on the whole book. We reformulated the sentence in the article in case it was unclear.
>
>
> > The hypothesis on line 406 should be tested empirically.
>
> We agree that this in an interesting question. Since there are more retrieval candidates in the whole book, some heuristics might need to retrieve more sentences in the book than in the current chapter to find relevant candidates. This is something we are currently exploring by performing more experiments, as these are straightforward.
>
>
> Please feel free to let us know if you have further questions.

---

### Official Review · Reviewer_SivA · 2023-08-05

**Soundness:** 3

**Excitement:**

3: Ambivalent: It has merits (e.g., it reports state-of-the-art results, the idea is nice), but there are key weaknesses (e.g., it describes incremental work), and it can significantly benefit from another round of revision. However, I won't object to accepting it if my co-reviewers champion it.

**Paper Topic And Main Contributions:**

The paper proposes a neural retriever that is trained on the synthetic dataset using Alpaca. The retriever could find relevant sentences from long novels to help the NER task. Specifically, the authors employ Vicuna to generate positive retrieving pairs through prompt instructions and use heuristic methods to form negative pairs. Then, the synthetic data is used to train the retriever, which could rank sentences in a small candidate set that is obtained from simple heuristic pre-retrieval. The top-k sentences are concatenated with the input sentence to finish the NER task. Experimental results show that the proposed method could outperform the unsupervised retrievers.

**Reasons To Accept:**

1.	The paper is well-written and easy to understand and follow.
2.	The idea of using LLM to generate training data is well-motivated and is valuable for further exploration in many tasks.
3.	Experiments show the proposed approach can achieve better results than traditional methods.

**Reasons To Reject:**

1.	The proposed method employs the neural retriever as a re-ranker to find top-k sentences from a candidate set. However, all the baselines have only a retrieval process without re-ranking. I suggest adding baselines that are equipped with a re-ranking stage, e.g., BM25 + TextRank.
2.	Since Vicuna could generate positive examples for building synthetic data, why cannot we just use the generated positive examples to augment NER tasks instead of retrieving them from a novel?
3.	In the experiments, only BERT-base model is evaluated, which might have limited performance in NER tasks and result in a performance boost when augmented with retrieved sentences. I would like to know whether the performance can still be improved when a more powerful model is employed as the backbone, e.g., BERT-large or RoBERTa. As a result, more comparison analyses should be provided to probe the effectiveness of the approach.

**Reproducibility:**

4: Could mostly reproduce the results, but there may be some variation because of sample variance or minor variations in their interpretation of the protocol or method.

**Reviewer Confidence:**

4: Quite sure. I tried to check the important points carefully. It's unlikely, though conceivable, that I missed something that should affect my ratings.

---

> ### Author Rebuttal · Authors · 2023-08-29
>
> Dear reviewer,
>
> Thank you for your review! We tried to answer to your remarks as best as we could below:
>
> > The proposed method employs the neural retriever as a re-ranker to find top-k sentences from a candidate set. However, all the baselines have only a retrieval process without re-ranking. I suggest adding baselines that are equipped with a re-ranking stage, e.g., BM25 + TextRank.
>
> We performed additional experiments on other baselines to answer this concern. We compared our re-ranker to a pretrained MonoBERT re-ranker (Nogueira and Cho, 2019). We chose MonoBERT since we wanted to compare to a model with a similar architecture in order to study the difference of performance caused by the training dataset. Still, note that the version of MonoBERT we employed is based on BERT large (huggingface checkpoint: `castorini/monobert-large-msmarco`), while our model is based on BERT base.
>
> We experimented with two variants of a MonoBERT based system:
>
> - bm25 -> MonoBERT, to replicate a classic BM25 + reranker baseline
> - [bm25, samenoun, surrounding] -> MonoBERT with $n=8$, to compare with the neural setup of our article (as in paragraph Inference of Section 4.1)
>
> We obtained the following mean results on 3 runs (F1-score):
>
> | System             /                 k    |     1 |     2 |     3 |     4 |     5 |     6 |     7 |     8 |
> | ----------------------------------------- | ----- | ----- | ----- | ----- | ----- | ----- | ----- | ----- |
> | bm25 -> MonoBERT                          | 92.18 | 91.91 | 91.93 | 91.98 | 91.90 | 91.58 | 91.65 | 91.44 |
> | [bm25, samenoun, surrounding] -> MonoBERT | 92.23 | 92.16 | 92.27 | 92.20 | 92.16 | 91.87 | 91.56 | 91.47 |
> | our                                       | 92.34 | 92.30 | 92.45 | 92.36 | 92.30 | 92.19 | 92.10 | 91.72 |
>
> Overall, our model obtains better performance for all values of $k$, even with MonoBERT having more parameters than our model. Our hypothesis is that the performance differences come from our retrieval dataset being specifically suited for NER and for the domain at hand. We will include these results in a revised version of the article.
>
>
> > Since Vicuna could generate positive examples for building synthetic data, why cannot we just use the generated positive examples to augment NER tasks instead of retrieving them from a novel?
>
> As mentioned in the article, we use the Alpaca LLM and not Vicuna, but your question still stands. This generation cannot be done at inference time because it requires the NER label of the entity to be correct. Since this label is unknown at inference time, a LLM could generate a context that mention the entity but resolves it to an incorrect type regarding the current novel. This is why we take advantage of the training set, which is labeled, to generate a context retrieval training dataset that we then use to train a re-ranker.
>
>
> > In the experiments, only BERT-base model is evaluated, which might have limited performance in NER tasks and result in a performance boost when augmented with retrieved sentences. I would like to know whether the performance can still be improved when a more powerful model is employed as the backbone, e.g., BERT-large or RoBERTa. As a result, more comparison analyses should be provided to probe the effectiveness of the approach.
>
> This is a good point, and we are currently working on an experiment to include results with BERT-large in the article.
>
>
> Please feel free to let us know if you have further questions.

---

### Official Review · Reviewer_1ozW · 2023-08-11

**Typos Grammar Style And Presentation Improvements:** 1) The authors should include an ethi…
**Soundness:** 4

**Excitement:**

3: Ambivalent: It has merits (e.g., it reports state-of-the-art results, the idea is nice), but there are key weaknesses (e.g., it describes incremental work), and it can significantly benefit from another round of revision. However, I won't object to accepting it if my co-reviewers champion it.

**Justification For Ethical Concerns:**

-

**Missing References:**

There should be another subsection in the Related works section to describe other re-rankers like T5 re-ranker etc.

T5 re-ranker reference:
@misc{hui2022retrieval,
      title={Retrieval Augmentation for T5 Re-ranker using External Sources},
      author={Kai Hui and Tao Chen and Zhen Qin and Honglei Zhuang and Fernando Diaz and Mike Bendersky and Don Metzler},
      year={2022},
      eprint={2210.05145},
      archivePrefix={arXiv},
      primaryClass={cs.IR}
}

The authors should also discuss about this work,
@article{Chawla2021ImprovingTP,
  title={Improving the performance of Transformer Context Encoders for NER},
  author={Avi Chawla and Nidhi Mulay and Vikas Bishnoi and Gaurav Dhama},
  journal={2021 IEEE 24th International Conference on Information Fusion (FUSION)},
  year={2021},
  pages={1-8},
  url={https://api.semanticscholar.org/CorpusID:244841717}
}

**Paper Topic And Main Contributions:**

The paper proposes a way to do supervised training of retrievers (pre-trained transformer-based) using relevant context from long documents for the NER task. Their main contributions are :
1) Proposed to generate a synthetic context retrieval training dataset using Alpaca.
2) Proposed to use this synthetic dataset for supervised training of neural context retriever based on the BERT model.
3) The authors showed that their retriever outperforms other unsupervised retrievers like BM25.

**Questions For The Authors:**

1) Why is there no comparison with other retrieval systems, like using a simple BM25 + T5 re-ranker or unsupervised retrieval systems?
2) What model is used for the final NER task? Is it also the BERT model?
3) Why did you write a long paper when you have to use up two pages for simple examples or redundant graphs? Even Amalvy et al. 2023 is a short paper.

**Reasons To Accept:**

1) Novel idea of training a supervised context retriever for NER using synthetic data that the authors generated.
2) The authors showed that their retriever outperforms other unsupervised retrievers like BM25.

**Reasons To Reject:**

1) The paper seems to simply fine tuned a BERT model and used it as a re-ranker. The authors didn't even compare their results with other re-ranker models. (Update: authors compared the results with mono-BERT and T5)
2) This paper should be written as a short paper only, the authors appears to have included example tables and redundant graphs on page 3 and 7 so as to meet the long paper requirements.
3) As the authors are using a simple retrieval heuristics and then using their retriever as a re-ranker it will always be dependent on heuristics first and hence it is essentially a simple retriever plus re-ranker system like BM25+T5 re-ranker.

**Reproducibility:**

4: Could mostly reproduce the results, but there may be some variation because of sample variance or minor variations in their interpretation of the protocol or method.

**Reviewer Confidence:**

4: Quite sure. I tried to check the important points carefully. It's unlikely, though conceivable, that I missed something that should affect my ratings.

---

> ### Author Rebuttal · Authors · 2023-08-29
>
> Dear reviewer,
>
> Thank you for your review! We tried to answer to your remarks as best as we could below:
>
> > Q1: Why is there no comparison with other retrieval systems, like using a simple BM25 + T5 re-ranker or unsupervised retrieval systems?
>
> While we included some baselines in our article, we performed additional experiments to address this concern. We compared our re-ranker to a pretrained MonoBERT re-ranker (Nogueira and Cho, 2019). We chose MonoBERT since we wanted to compare to a model with a similar architecture in order to study the difference of performance caused by the training dataset. Still, note that the version of MonoBERT we employed is based on BERT large (huggingface checkpoint: `castorini/monobert-large-msmarco`), while our model is based on BERT base.
>
> We experimented with two variants of a MonoBERT based system:
>
> - bm25 -> MonoBERT, to replicate a classic BM25 + reranker baseline
> - [bm25, samenoun, surrounding] -> MonoBERT with $n=8$, to compare with the neural setup of our article (as in paragraph Inference of Section 4.1)
>
> We obtained the following mean results on 3 runs (F1-score):
>
> | System             /                 k    |     1 |     2 |     3 |     4 |     5 |     6 |     7 |     8 |
> | ----------------------------------------- | ----- | ----- | ----- | ----- | ----- | ----- | ----- | ----- |
> | bm25 -> MonoBERT                          | 92.18 | 91.91 | 91.93 | 91.98 | 91.90 | 91.58 | 91.65 | 91.44 |
> | [bm25, samenoun, surrounding] -> MonoBERT | 92.23 | 92.16 | 92.27 | 92.20 | 92.16 | 91.87 | 91.56 | 91.47 |
> | our                                       | 92.34 | 92.30 | 92.45 | 92.36 | 92.30 | 92.19 | 92.10 | 91.72 |
>
> Overall, our model obtains better performance for all values of $k$, even with MonoBERT having more parameters than our model. Our hypothesis is that the performance differences come from our retrieval dataset being specifically suited for NER and for the domain at hand. We will include these results in the a revised version of the article.
>
>
> > Q2: What model is used for the final NER task? Is it also the BERT model?
>
> Our model consists of a BERT encoder followed by a classification layer that we finetune on the NER task (see Section 4.2).
>
>
> > Q3: Why did you write a long paper when you have to use up two pages for simple examples or redundant graphs? Even Amalvy et al. 2023 is a short paper.
>
> We were not trying to transform a short paper into a long paper by adding redundant graphs. We chose to include example tables because we think it benefits the reader's comprehension of the article, since the dataset generation is not straightforward. Additionally, removing these figures would not transform the current paper into a short one.
>
>
> > There should be another subsection in the Related works section to describe other re-rankers like T5 re-ranker etc.
>
> We agree this warrants a subsection in the related works, and we will include it in a revised version of the article.
>
>
> > The authors should also discuss about this work: Improving the performance of Transformer Context Encoders for NER
>
> As we understand it, this article proposes several enhancements to NER systems (data augmentation, usage of a character-level encoder), but is not concerned with retrieval or the influence of long-range context in NER.
>
>
> > The authors should include an ethics statement section in the paper.
>
> To our understanding the ethics statement is not mandatory, and we did not see a reason to include one. Do you think there are specific concerns about our work that we should address in such a section?
>
>
> > The authors should explain explicitly that "no retrieval" means the NER model is not given any context and that it is a simple baseline (line 261).
>
> We corrected the article and added it explicitly.
>
>
> Please feel free to let us know if you have further questions.

---

### Official Review · Reviewer_YYw4 · 2023-08-14

**Soundness:** 2

**Excitement:**

3: Ambivalent: It has merits (e.g., it reports state-of-the-art results, the idea is nice), but there are key weaknesses (e.g., it describes incremental work), and it can significantly benefit from another round of revision. However, I won't object to accepting it if my co-reviewers champion it.

**Paper Topic And Main Contributions:**

This research focuses on retrieving context for the NER task. In particular, it leverages the large language (LLM) model (i.e., alpaca) to generate supervision to train the context retrieval model. The proposed neural context retriever with several simple baselines.


**Questions For The Authors:**

Do you plan to share the proposed dataset with the public? If so, what would be the license of your dataset?

**Reasons To Accept:**

Data augmentation with LLM is an interesting/promising direction that can be applied to many NLP downstream. Good to see research that applies it to the NER domain.


**Reasons To Reject:**

The authors propose a method to generate a trainset from LLM; however, there is no analysis of the quality of the generated samples.

The proposed model outperformed the baselines; however, these baselines are not strong enough to justify the superiority of the proposed model. The author should compare the final NER performance with other methods to claim that the proposed method significantly improves performance in this area.


**Reproducibility:**

2: Would be hard pressed to reproduce the results. The contribution depends on data that are simply not available outside the author's institution or consortium; not enough details are provided.

**Reviewer Confidence:**

3: Pretty sure, but there's a chance I missed something. Although I have a good feel for this area in general, I did not carefully check the paper's details, e.g., the math, experimental design, or novelty.

---

> ### Author Rebuttal · Authors · 2023-08-29
>
> Dear reviewer,
>
> Thank you for your review! We tried to answer to your remarks as best as we could below:
>
> > Do you plan to share the proposed dataset with the public? If so, what would be the license of your dataset?
>
> As indicated in the article, we release the generated datasets. You can find these datasets in the anonymized repository under _runs/gen/genv3_ (Alpaca 7b) and _runs/gen/genv3\_13b_ (Alpaca 13b). Datasets are under CC BY-NC 4.0 license. As pointed by reviewer NDUN, we also released code and instructions to replicate all of our experiments.
>
>
> > The proposed model outperformed the baselines; however, these baselines are not strong enough to justify the superiority of the proposed model. The author should compare the final NER performance with other methods to claim that the proposed method significantly improves performance in this area.
>
> We performed additional experiments on other baselines to answer this concern. We compared our re-ranker to a pretrained MonoBERT re-ranker (Nogueira and Cho, 2019). We chose MonoBERT since we wanted to compare to a model with a similar architecture in order to study the difference of performance caused by the training dataset. Still, note that the version of MonoBERT we employed is based on BERT large (huggingface checkpoint: `castorini/monobert-large-msmarco`), while our model is based on BERT base.
>
> We experimented with two variants of a MonoBERT based system:
>
> - bm25 -> MonoBERT, to replicate a classic BM25 + reranker baseline
> - [bm25, samenoun, surrounding] -> MonoBERT with $n=8$, to compare with the neural setup of our article (as in paragraph Inference of Section 4.1)
>
> We obtained the following mean results on 3 runs (F1-score):
>
> | System             /                 k    |     1 |     2 |     3 |     4 |     5 |     6 |     7 |     8 |
> | ----------------------------------------- | ----- | ----- | ----- | ----- | ----- | ----- | ----- | ----- |
> | bm25 -> MonoBERT                          | 92.18 | 91.91 | 91.93 | 91.98 | 91.90 | 91.58 | 91.65 | 91.44 |
> | [bm25, samenoun, surrounding] -> MonoBERT | 92.23 | 92.16 | 92.27 | 92.20 | 92.16 | 91.87 | 91.56 | 91.47 |
> | our                                       | 92.34 | 92.30 | 92.45 | 92.36 | 92.30 | 92.19 | 92.10 | 91.72 |
>
> Overall, our model obtains better performance for all values of $k$, even with MonoBERT having more parameters than our model. Our hypothesis is that the performance differences come from our retrieval dataset being specifically suited for NER and for the domain at hand. We will include these results in a revised version of the article.
>
>
> Please feel free to let us know if you have further questions.

---

### Meta-Review · Area_Chair_FJ6k · 2023-09-22

**Recommendation:** 3

**Metareview:**

***Quality***: Reviewers largely agreed that this work investigates a promising research topic and makes research contributions. Several reviews noted that the experimental evaluation did not sufficiently ablate performance improvements to differentiate improvements from data augmentation versus re-ranking. The authors performed additional experiments during the rebuttal period to address these issues. The authors provided sufficient materials to allow reproducibility of their findings.

***Clarity***: Reviewers found the ideas to be clear, although suggested some details and discussion be added to improve clarity.

***Originality***: The core idea of this paper (data augmentation using LLMs) is an area of active research, and is not novel. The application to contextual NER is an original application of this research approach.

***Significance***: Improvements from the proposed method on the NER task investigated were small, but the task area is mature and small improvements are still meaningful. This paper is of interest to the EMNLP community and provides a useful contribution. Overall, the paper did not generate strong excitement from the reviewers, so the impact to the community may be small.

***Pros***:
* Promising research direction for contextual NER
* Clear ideas and a novel research contribution
* Reproducible experiments supporting the claim

***Cons***:
* Additional experiments necessary to more clearly support the method
* Extending to other datasets with different NER tags may be difficult

---

### Decision · Program_Chairs · 2023-10-07

**Decision:**

Accept-Main

**Comment:**

***Quality***: Reviewers largely agreed that this work investigates a promising research topic and makes research contributions. Several reviews noted that the experimental evaluation did not sufficiently ablate performance improvements to differentiate improvements from data augmentation versus re-ranking. The authors performed additional experiments during the rebuttal period to address these issues. The authors provided sufficient materials to allow reproducibility of their findings.

***Clarity***: Reviewers found the ideas to be clear, although suggested some details and discussion be added to improve clarity.

***Originality***: The core idea of this paper (data augmentation using LLMs) is an area of active research, and is not novel. The application to contextual NER is an original application of this research approach.

***Significance***: Improvements from the proposed method on the NER task investigated were small, but the task area is mature and small improvements are still meaningful. This paper is of interest to the EMNLP community and provides a useful contribution. Overall, the paper did not generate strong excitement from the reviewers, so the impact to the community may be small.

***Pros***:
* Promising research direction for contextual NER
* Clear ideas and a novel research contribution
* Reproducible experiments supporting the claim

***Cons***:
* Additional experiments necessary to more clearly support the method
* Extending to other datasets with different NER tags may be difficult